# Lessons from piloting and scaling a real-time DHIS2 based treatment reporting tool for mass drug administration in Nigeria

Martins Imhansoloeva[1]*, Christian Nwosu[1], Omosefe Osinoiki[1], Sunday Isiyaku[1], Chukwuma Anyaike[2], Perpetua Amodu-Agbi[2], Ununumah Egbelu[2], Augustine Nwoye[2], Peter Oyinloye[3], Amin Umar Abdurahman[4], Ifeoma Otiji[5], Lazarus Nweke[6], Joseph Kumbur[7], Rinpan Ishaya[8], Ruth Dixon[9], Sarah Bartlett[9]

**1** Sightsavers, Nigeria Country Office, Abuja, Nigeria, **2** Neglected Tropical Diseases Division, Federal Ministry of Health, Abuja, Nigeria, **3** Neglected Tropical Diseases Unit, Kwara State Ministry of Health, Ilorin, Kwara State, Nigeria, **4** Neglected Tropical Diseases Unit, Jigawa State Ministry of Health, Dutse, Jigawa State, Nigeria, **5** Neglected Tropical Diseases Unit, Enugu State Ministry of Health, Enugu, Enugu State, Nigeria, **6** The Carter Centre, Enugu, Enugu, Nigeria, **7** Christian Blind Mission, Dutse, Jigawa State, Nigeria, **8** Health and Development Support Programme, Dutse, Jigawa State, Nigeria, **9** Sightsavers, Haywards Heath, West Sussex, United Kingdom

* mimhansoloeva@sightsavers.org

## Abstract

### Background

Mass drug administration (MDA) is the main intervention strategy for the elimination of several neglected tropical diseases (NTDs). In many endemic settings, monitoring and collation of MDA treatment data are conducted through paper-based forms and Excel-based spreadsheets. These methods are often slow, prone to errors and do not facilitate timely evidence-based decision making during and after MDA campaigns. The Nigerian National NTD programme and Sightsavers, developed a DHIS2 based platform for real-time collection, monitoring, and reporting of MDA treatment data. We piloted and scaled this DHIS2-based platform, monitoring the data quality, access, government ownership and utility of data for programmatic action at all levels.

### Methods

Three study areas (Jigawa, Enugu and Kwara States), with upcoming MDA campaigns were selected based on geographic spread and model of non-governmental development organisation (NGDO) implementing partner support. Following a pilot in Jigawa State, the DHIS2 platform was scaled-up across all three study areas, alongside the existing Excel-based systems. Programmatic data routinely collected via the two platforms were compared. Instances of data entry and access were monitored via the platform's metadata and a monitored helpline. Data was collected from participants through a self-administered questionnaire, field diaries and focus group

**Data availability statement:** All relevant data are within the manuscript and its Supporting Information files.

**Funding:** This work was supported by The Task Force for Global Health through the NTD Support Center (#232851287 to MA, RD, OO, SI, CN and SB at Sightsavers). The funders had no role in study design, data collection and analysis, decision to publish, or preparation of the manuscript.

**Competing interests:** The authors have declared that no competing interests exist.

discussions/key informant interviews. Quantitative data was analysed using Stata analytical software, while qualitative data was thematically analysed.

## Results

There was increased access and use of data at all levels within the DHIS2 system along with improved perceptions of government ownership of the data. Participants reported the ability to address errors and improve decision-making during campaigns as significant benefits of the platform. Scaling up DHIS2 was feasible, and similar benefits were observed in all the models of NGDO partner assistance.

## Conclusion

The DHIS2 platform enhanced all components of ownership, as well as demonstrated ability to be replicated in different settings. However, operating models, cultural contexts, and technical capacities across the diverse locations need to be considered when scaling up the platform.

### Author summary

Mass drug administration (MDA) campaigns for neglected tropical diseases (NTDs) often rely on paper-based forms and Excel spreadsheets for data collation, monitoring, and reporting. These traditional methods are slow, error prone, and do not support timely, evidence-based decision-making to enhance MDA campaigns. The need for accurate, real-time reporting of MDA data during implementation remains critical for many national NTD programs. We piloted a new DHIS2-based platform co-developed by the national NTD programme and Sightsavers for reporting MDA data in Nigeria. This platform was subsequently scaled up in three different programmatic settings to assess its feasibility and explore the challenges of implementing the platform. We also assessed the potential of the platform to improve government ownership among program implementers. Findings indicates that the platform enhanced data access and ownership across the different levels of the health system. Program implementers also found the platform highly beneficial, and there was notable enthusiasm for its use, as it facilitated quicker decision-making during and after the MDA campaign. Key considerations regarding partner support, technological capabilities, internet penetration especially in rural areas, among others, are suggested as issues that must be addressed when scaling the platform more widely within a national program.

## 1. Background

Neglected tropical diseases (NTDs) are a diverse group of communicable diseases affecting 1.5 billion people worldwide [1]. These diseases

disproportionately impact impoverished communities, causing significant morbidity and mortality. In sub-Saharan Africa, Nigeria bears the highest burden of NTDs, accounting for 25% of the entire region's NTD burden, with over 122 million people at risk of developing one or more of these NTDs [2]. For five NTDs, known as preventative chemotherapy (PC) NTDs - lymphatic filariasis, onchocerciasis, soil-transmitted helminthiasis, schistosomiasis, and trachoma, the current WHO recommended approach to achieve elimination is regular annual or bi-annual mass drug administration (MDA) [3].

In Nigeria, the PC- NTDs are targeted for elimination by the year 2030 in line with the WHO 2030 NTD road map [2], and improving MDA coverage for NTDs is highlighted in the national health policy [4]. Implementation of MDAs is led by the Federal Ministry of Health (FMoH) through its national and/or State (sub-national level) NTD programmes with support from non-governmental development organisation (NGDO) implementing partners. Sightsavers is one such NGDO implementation partners and have been working with FMoH since 1953, co-implementing programmes that have delivered more than 590 million PC-NTD treatments across seven states [5]. For lymphatic filariasis, and onchocerciasis, MDA is implemented at the community level via household distribution to the whole population of over 5 years [6], while schistosomiasis and soil-transmitted helminthiasis MDAs are delivered mainly to school aged children (SAC) via household or school based campaigns [7,8].

Nigeria adopted a DHIS2 based health management information system (HMIS) in 2010 [9]. The HMIS is used for reporting, analysing, and dissemination of routine data for several health programmes including polio, malaria, tuberculosis, and HIV/AIDs amongst others [10]. In common with many Ministries of Health globally, the FMoH reports on PC-NTD MDAs by printing paper treatment registers which are used by community drug distributors (CDDs) to record treatment information as it occurs [11]. Various paper summary sheets are then used to collate data from communities, and entered into an electronic spreadsheet. The electronic data are aggregated through different administrative levels up to the federal level. While a paper based aggregating system connects well with the community level at scale, these systems are often linked with untimely data reporting and poor data quality.

In Nigeria, the reliance on paper based system in combination with the complex partner landscape leads to data inconsistencies and fragmentation, in turn affecting timeliness, quality, and availability of comprehensive treatment data both during and after the MDA campaign [12]. There is also little to no visibility on progress and on issues during campaigns. Additionally, sub-district drug stock is not tracked and there are no standard processes for drug re-collection or redistribution due to the complexity of such a process. Although the FMoH technically owns the data, the lack of centralised storage and control hampers the use of their data for timely programmatic decision making in response to emerging issues, strategic planning and optimal resource allocation [12,13].

Between 2019 and 2021, FMoH and Sightsavers developed, piloted and scaled an electronic data collection platform to replace the existing spreadsheet system in Nigeria. The platform was based on the DHIS2, similar to that developed for real-time monitoring for trachoma MDA campaigns in Zimbabwe [14]. The platform was designed for easy integration into the HMIS as and when determined by FMoH. The platform was designed to generate accessible, clean and complete MDA data in real time during ongoing MDAs as well as to centralise both historic and future MDA treatment data. It was intended that in addition to enabling programmatic adjustments during active campaigns, the platform would enhance long term Ministry ownership over the data, contribute to monitoring trends and streamline reporting processes to the WHO.

This paper presents the results from piloting and a heavily monitored first stage of scaling of the new platform in different MDA settings in Nigeria. The objectives are: to assess, in different contexts, the platform's ability to generate timely, high quality and accessible data; to determine if and how that data was or could be used for programmatic action; to understand any impact on the sense of government ownership and to identify barriers, solutions and enablers to inform ongoing roll out.

## 2. Methods

### 2.1. Ethics statement

Ethical approval was sought from the National Health Research Ethics Committee (NHREC) at the Federal Ministry of Health, Abuja. Due to integration with programmatic activities, it was deemed eligible for an ethical waiver with Health Research Ethics Committee (HREC) assigned number: NHREC/01/01/2007. During the Covid-19 pandemic, adjustments were made to ensure that all research activities adhered to the regulations set out by the Nigerian Centre for Disease Control.

### 2.2. Selection of study areas

Using the national schedule of upcoming MDAs, three States (Jigawa, Enugu and Kwara) were purposively selected for piloting and initial scale up such that there was a diversity of type of MDA, technical support partner and model, and geographic region of Nigeria (Table 1). Three local government areas (LGAs) were selected from Jigawa State (with the soonest upcoming MDA) and then prioritized for an initial pilot which took place in March 2020. Two LGAs selected from two additional states followed in a fuller scale-up between December 2020 and March 2021. The LGAs were chosen as the primary unit, being the implementing units and for which MDA treatment data are aggregated and reported to the national level.

### 2.3. Piloting and Scaling-up of the DHIS2 Platform

The DHIS2 platform was rolled out in the selected States during their respective MDAs. Prior to the MDA campaign, community forms were configured and targets pre-populated into the DHIS2 platform. A "Command Centre" – a group put together at a central location to monitor treatment data as the MDA campaign took place - was set up and incorporated as part of the process. Training was conducted for LGA monitoring and evaluation (M&E) officers and Command Centre personnel (including state NTD coordinator, assistant NTD coordinator, M&E manager, logistics manager, and HMIS manager). During the MDA campaign, the LGA M&E teams entered treatment data into the DHIS2 platform, which was fed into the dashboard. This was supported by dashboard monitoring, and a toll-free helpline by the Command Centre to provide feedback to the data entry teams. Post-campaign, review meetings assessed both implementation processes and outcomes, as well as informed future planning (Fig 1).

**Table 1. Roll out schedule of the MDA in the pilot and scale-up states.**

| Background | | | | | | DHIS2 system roll out | | |
|---|---|---|---|---|---|---|---|---|
| Type of MDA | State/ Region | Technical Assistance Partner | # LGAs in the MDA | Dates of MDA implementation | # treatment targeted | # LGAs | Date of LGA level training | # people trained |
| Onchocerciasis/Lymphatic filariasis | Jigawa/ NW | CBM and HANDS | 14 | February – March 2020 | 2,701,991 | 14 | 7th February 2020 | 71 |
| Onchocerciasis/Lymphatic filariasis | Enugu/ SE | The Carter Center | 4 | Dec 2020 – Jan 2021 | 995,540 | 4 | 1st - 5th September 2020; 14th November* | 21 |
| Onchocerciasis/Lymphatic filariasis | Kwara/ SW | Sightsavers | 16 | March 2021 | 3,350,287 | 16 | 3rd – 5th March, 2021 | 76 |
| Schistosomiasis/ soil-transmitted helminthiasis | Jigawa/ NW | CBM and HANDS | 25 | March 2021 | 3,172,028 | 25 | 25th February, 2021 | 106 |

*Refresher training

NW – Northwest Nigeria; SE – Southeast Nigeria; SW – Southwest Nigeria; LGA – local government area; MDA – Mass Drug Administration; CBM – Christian Blind Mission; HANDs - Health and Development Support Programme

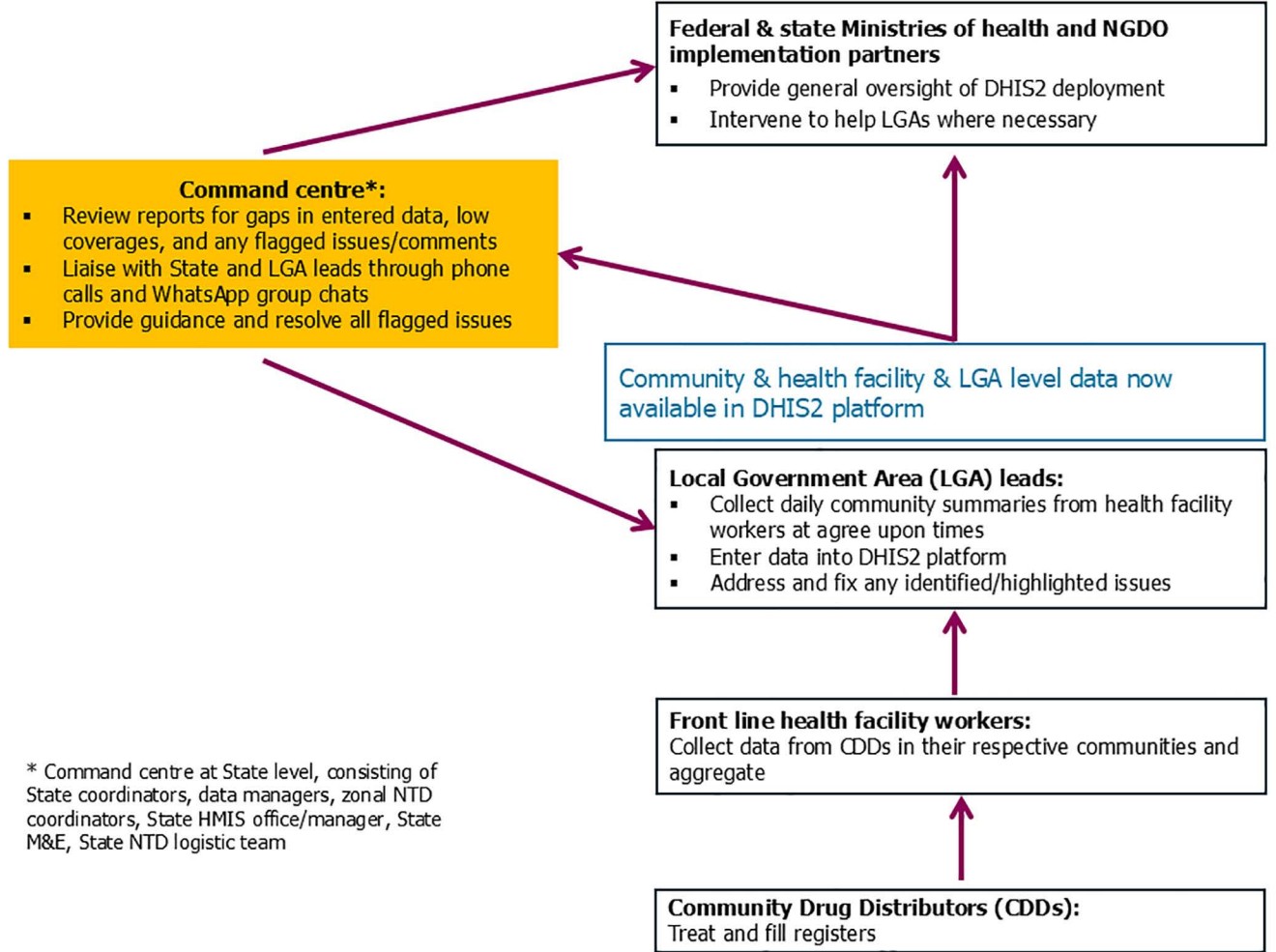

**Fig 1. Roles and data flow using the DHIS2 platform during an MDA campaign in Nigeria.**

Trainings were initially run by Sightsavers during the pilot, while subsequent trainings were facilitated by FMoH with technical support from Sightsavers. During both pilot and scaleup MDAs, the DHIS2 platform was used alongside the paper-based reporting.

### 2.4. Data collection procedure

Data collection for the process evaluation of the DHIS2 platform was done using different methods including meta-data analysis, helpline monitoring logs, coverage evaluation survey, field diaries, focus group discussions and key informant interviews. We assessed and the DHIS2 platform's ability to improve on the key study outcomes: data access; quality of data, accuracy of treatment coverage, sense of ownership. The feasibility and acceptability of the DHIS2 platform was also assessed.

**2.4.1. Analysis *of* DHIS2 Meta-data.** The metadata from DHIS2 includes a list of all created usernames and their corresponding user assignments, along with a record of system access instances, detailing the date, time, and module or

process undertaken. Following each MDA implementation and the completion of reporting into DHIS2, the metadata was exported to Excel and imported into STATA for analysis.

**2.4.2. Helpline monitoring logs.** A helpline monitoring sheet was kept as a booklet and used to record all communication from LGAs to the Command Centre via the toll-free lines so that LGA issues could be recorded, categorised, and reviewed. The monitoring sheet had 14 columns which included reference, location date, time, call answered by, name and position of person reporting the issue, LGA reporting from, issue reported, issue code, what was done to respond to issue reported, remarks (on if issues was successfully resolved, pending or unresolved), call length, incoming phone number if available, follow-up action/s needed to be taken (who needs to be contacted, what further actions are needed). A total of 73 call records (24 in Enugu, 19 in Kwara and 30 in Jigawa) were documented during the scale-up implementation and evaluation.

**2.4.3. Coverage Evaluation Surveys (CES).** We conducted coverage evaluation surveys (CES) following MDA exercises to assess treatment coverage and compare these results with coverage estimates reported through the DHIS2 platform. The objective was to evaluate the level of agreement between the CES-derived coverage and the administrative coverage data from DHIS2, and to identify any discrepancies between the two sources. Sample size calculation, selection of sub-units, segmentation and data analysis followed standard World Health Organization guidance on preventive chemotherapy coverage surveys [15]. For each survey, sample size was calculated, and communities were selected using the WHO Coverage Survey Builder (CSB). Within each community cluster, a fixed number of households were randomly selected, and following verbal consent, all household members were enumerated and asked if they had received and swallowed the medicine, along with basic demographic details. During the pilot, all three evaluation LGAs were included. In the scale-up phase, two LGAs were randomly selected in each State, except for Jigawa where four were selected; two were done remotely, and two were done with a centralised team. Data was collected on Android smartphones using CommCare data collection software and exported into STATA for analysis.

**2.4.4. Field diaries.** Participants were trained and asked to keep field dairies documenting the activities, experiences, and events around the use of the DHIS2 platform as it was being used during the MDA campaign. These diaries included an activity log and recorded both positive and negative experiences with the platform, such as data entry, viewing the dashboard, interacting with the Command Centre via the hotline, and responding to technical and programme issues. Field diaries were maintained in randomly selected LGAs, with participants selected based on their roles in the Ministry and their involvement in the DHIS2 real-time reporting roll out for MDA. Between 10–14 participants from FMoH, State Ministries of Health (SMoHs) and LGAs maintained field diaries for 5 days (Jigawa pilot) or 10 days (Enugu, Kwara, Jigawa scale-up) of the MDA campaign.

During the pilot stage, field diaries were collected from 10 participants using the WhatsApp voice notes feature, while 14 participants recorded field diaries during the scale up. In both Enugu and Kwara, field diaries were collected from 12 participants respectively. Data collectors documented and sent between 2–5 voice notes daily to a dedicated WhatsApp line, with activities also recorded on a paper matrix. In the scale phase, data collection occurred using a CommCare app on Android phones, integrating both the voice recording and activity logging. Data collection occurred concurrently with MDA implementation at the LGA level and was monitored by the SMoH and FMoH representatives.

**2.4.5. Self-administered questionnaire.** An assessment of ownership was conducted with government policy makers and implementers at the FMoH, SMoH and LGA levels at three points: 1) Before engaging with the DHIS2 system, reflecting on previous MDA ownership experience, 2) After the DHIS2 training, anticipating future ownership and 3) After the MDA rollout, reflecting on actual ownership experience. Elements of government ownership were explored and defined at a participatory workshop and assessed using collaboratively developed statements on a Likert scale (Table A in S1 File). The aim was to understand participants' satisfaction with government ownership of MDA data before and after the DHIS2 rollout (comparing scores 1 and 3) and to gauge perceptions vs. reality (comparing scores 2 and 3). Scores

were computed for three components of government ownership, with a maximum of 15 points per component and 45 points overall. A total of 129 participants from the three states (26, 49 and 54 from Enugu, Kwara and Jigawa respectively) responded to the questionnaire.

**2.4.6. Key informant interviews and focus group discussions.** Nine key information interviews (KIIs) focusing on ownership were conducted with NTD programme representatives from the SMoH, LGA NTD Units and the NDGO partners. One LGA NTD focal person and one LGA M&E officer were purposively selected per State, while one person (State coordinator or the Data Manager) was purposively selected per State. Interviews were facilitated using an interview guide containing a series of questions structured around three broad themes of ownership identified in the participatory workshop at the start of the research: data access, data use and data control.

Three focus group discussions (FGDs) were conducted with programme implementers to identify ways in which the DHIS2 platform has enhanced government ownership with respect to the MDA. One FGD was conducted with the FMoH team, while 2 FGDs were conducted with the scale-up States split into two groups. Participants for the FGDs ranged between 3–8, and discussions were facilitated using an FGD guide which focused on the three areas: 1) ownership 2) improvements/solutions suggested or required and 3) barriers to scale up.

## 2.5. Data analysis

We analysed the meta-data by calculating the following indicators: proportion of persons who accessed the system (after been configured and assigned access to the platform); proportion of access events per administration level; and proportion of different activities undertaken by each administration level.

Indicators from call monitoring logs were also calculated for each State, and included the total number of calls received, the proportion/status of issues reported grouped into five categories (user login, community listing, drug inventory, treatment coverage, others) and the status to show if the issue was resolved or not.

Therapeutic coverage from the CES was calculated for each LGA as the number of people who swallowed the medicine (treated) divided by the total population in the surveyed households. We also calculated availability (communities where >80% of households were offered the medicines), accessibility (proportion of individuals offered the medicine), and effective coverage (proportion of individuals who swallowed the offered medicine, analogous to therapeutic coverage) based on the Tanahashi framework [16].

Questionnaire data was analysed by estimating the significant changes in mean scores using matched sample tests. Participants who did not complete both relevant time points were excluded from the analysis.

Field diary notes, FGDs and KIIs recordings were transcribed verbatim for analysis. Transcripts were read by different members of the research team and then thematically analysed based on the structural themes of ownership (access, utility, and control of data). Data analysis was performed using NVivo 12.

## 3. Results

### 3.1. DHIS2 meta-data analysis

Analysis of programme and meta-data analysis were conducted to determine the level of data access by the different states. A total of 228 persons were configured onto the DHIS2 platform across the three States – Enugu (26), Jigawa (114), Kwara (67) and FMoH (20). Of the 228 persons trained, 127 (56%) accessed the platform to perform any activity (data visualisation or data entry). This was higher in Enugu (93%) compared to Kwara 51% and Jigawa (46%).

Regarding the activities undertaken, LGA teams reported the highest incidence of events/person (351), compared to State (140 events/person) and FMoH (84 events/person). In Kwara State visualizing was the predominant activity (63.2%), while in both Jigawa and Enugu it was data entry (88.9% and 65.0% respectively). (Fig 2).

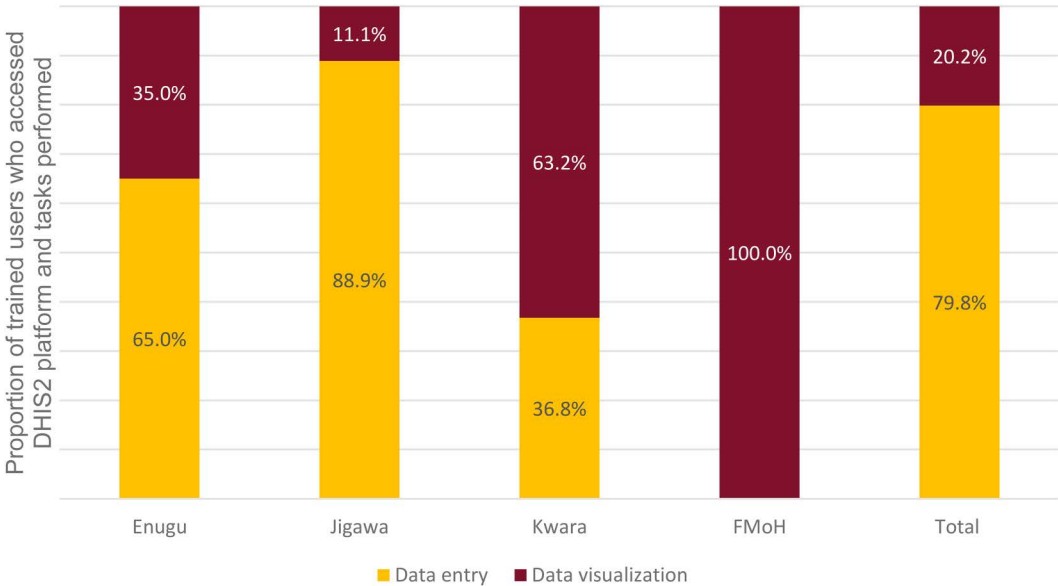

**Fig 2. Proportions of persons across the different states and FMoH accessing the DHIS2 platform and tasks performed.**

## 3.2. Helpline monitoring

Results of helpline monitoring logs showed 73 records; Enugu (24), Jigawa (30) and Kwara (19). More than one-third of the issues recorded related to "other issues" such as multiple data submission, difficulty in data retrieval, adjusting data and solving errors. This was followed by issues around drug inventory (21.9%), user login (19.2%), community listing and treatment coverage (12.3% respectively). The majority of the issues were successfully resolved during the calls in Kwara (75%) and Enugu (95%), and Jigawa States (100%). Table 2 shows the distribution of recorded issues reported by each State.

## 3.3. Treatment coverage data

Table 3 shows the treatment coverage estimates obtained from coverage evaluation surveys (CES), the programme spreadsheet and the DHIS2 system. Results showed that the coverage estimates from the two scale-up LGAs in Kwara and one scale-up LGA in Enugu were higher than those obtained from the CES, as opposed to the other LGAs which reported lower coverages in the DHIS2 compared to the CES. Similarly, apart from one of the pilot LGAs in Jigawa and one scale -up LGA in Enugu, most coverage estimates from the DHIS2 platform from the states, fell outside of the confidence intervals of the CES. Overall, there were inconsistencies in the results of treatment coverage obtained from the three sources.

**Table 2. Comparison of helpline monitoring data across the three scale-up States.**

| n (%) | User login | Community listing | Drug inventory | Treatment coverage | Others | Total |
|---|---|---|---|---|---|---|
| Enugu | 1 (4.2) | 1 (4.2) | 2 (8.3) | 10 (41.7) | 10 (41.7) | 24 (32.9) |
| Jigawa | 8 (26.7) | 2 (6.7) | 6 (20.0) | 7 (23.3) | 7 (23.3) | 30 (41.1) |
| Kwara | 5 (26.3) | 6 (31.6) | 8 (42.1) | 0.0 | 0.0 | 19 (26.0) |
| **Total** | **14 (19.2)** | **9 (12.3)** | **16 (21.9)** | **9 (12.3)** | **25 (34.2)** | **73 (100)** |

**Table 3. Comparison of treatment coverage estimates from multiple sources across the three scale-up States.**

| State | LGA | Source of treatment coverage | | Within CES 95% CI | Programme Estimate* |
|-------|-----|------------------------------|---|-------------------|---------------------|
| | | CES coverage (95% Confidence Interval - CI) | DHIS2 coverage | | |
| Jigawa (pilot) | Babura | 72.8 (71.2 – 74.4) | 61.0% | No | |
| | Sule Tankarkar | 85.7 (84.4 – 86.9) | 52.5% | No | |
| | Taura | 77.7 (76.0 – 79.2) | 63.4% | No | |
| Jigawa (scale) | Babura | 86.5 (71.7 – 94.1) | 81.7% | Yes | 89.9% |
| | Birnin Kudu | 89.8 (82.7 – 94.2) | 75.8% | No | 100% |
| Enugu (scale) | Nkanu East | 48.4 (40.8 – 56.1) | 61.3% | No | |
| | Isi Uzo | 62.2 (54.8 – 69.0) | 58.0% | Yes | |
| Kwara (scale) | Isin | 68.6 (61.0 – 75.3) | 84.0% | No | 80.0% |
| | Offa | 50.6 (37.7 – 63.4) | 83.9% | No | 79.3% |

*where available

### 3.4. Field diaries

Results from the field diaries provided a qualitative understanding of people's experiences around accessibility of data, use of data and quality of data.

**3.4.1. Accessibility of data.** Multiple experiences of accessing the DHIS2 system and the data were described by participants at LGA, State and Command Centre level. At the Command Centre and the State level participants felt pleased and happy with what they described as increased, faster and unrestricted data access, describing a sense of empowerment in their ability to check incoming data and identify issues. Such issues were with the data entry or mistaken counts (data quality) while others were focused on MDA itself such as MDA not occurring or medicine shortages.

*"I have been focusing on data review, pooling the data from the different organisations unit and running analysis to see if it looks good. So far it has been good I think I am seeing everything I want to see from the DHIS2 I am able to see the gap and all. So, I think I am excited"* (Federal, Jigawa Scale Up).

*"I now do it, within 10secs, I am happy….I set up the indicators in the pivot table and once I do that like for example, let me use yesterday, 15th, I came quickly and set up within 10secs, I was through with the pivot table immediately looked the community level;….I quickly noticed absence of data in one health post (ikpakpara HP) under okpu ward in isi-uzo LGA. To my greatest surprise, quickly followed it up, and there is no MDA going on in the 20 communities under the health post under Ikpakpara HP"* (State, Enugu Scale).

*"The good thing is that we are seeing at real time, we can quickly intervene, we are seeing all these things",* (Federal, Kwara scale)

*"This DHIS tool helps us to detect the village that has not submitted their data, and it helps us to detect the CDD or facility staff or LIHC that is not working. This helps us to be fast and up and doing. Everybody was on field to finish up their data. This tool, everybody loves it, I introduced the tool to my staff, to my facility staff and LIHC, they are all happy with the tool because you will see that if you don't submit your data, you will have, the column will be vacant, and it will be showing it to you so that everybody must be up and doing."* (LGA, Enugu scale)

**3.4.2. Infrastructure for using DHIS2 Platform.** Inadequate infrastructure was expressed as one of the impediments to accessing the DHIS2 platform. Persons in the Command Centre and at the State level reported

some difficulties with getting used to the system; they also reported access issues where the platform failed to upload and/or where internet data bundles provided were insufficient and had to be topped up or supplemented personally.

*"I used to find it not too easy, a little hard to set up my 3Ws for my pivot tables, setting the data elements, 3W such as the data, the period, where and the organization unit -I used to find it difficult to do it" (State, Enugu scale).*

*"The other thing is just the poor internet network it is not the fault of the DHIS2 platform. It is just that it slows down work in trying to do or to check or download and it is just annoying, and we are resolving that. Sometimes I use my wifi or my data, so it makes it move faster" (Federal, Kwara scale).*

*"I am still finding it very difficult to logon to the dashboard and it is really very annoying. Initially I thought it had something to do with the network, but I tried logging on to other (web) sites and other sites are opening just the dashboard refused to open. I even tried to do an alternate by coming in through the data entry coupon but even that, it did not open. I will just keep continuing but it is frustrating" (Federal, Jigawa pilot).*

At the data entry level (LGA), although users reported occasions or experiences of simple and easy access experiences which were "*with no more stress*" it was also described as frustrating as they found the system difficult to access, suffered from poor network coverage, power failure, lack of computer availability or being unable to correct errors that had been pointed out to them. Within the system a recurrent issue was the community listings where those doing the entry could not find the communities in the system in order to do the entry. LGA staff entering data could not change the lists themselves. This particular experience was shared by those at the Command Centre level who felt that the LGAs or States had not engaged fully when the community listings were defined leading to a lot of mismatching at a later point.

*"The problem which we are encountering during data entering is poor network service because sometimes the service is fluctuating, and we are unable to login to the DHIS2. The problem is coming from the service providers, and we are doing our best since morning we are able to put so many schools and we have many at our hand now" (LGA, Jigawa Scale)*

*"...Part of the things we observe today were issues around community listing.... You know, for the State the LGA will collect and send. And these are the communities and population we will create in the platform but on implementation, the whole scenario, will change, we start hearing we are looking for this community; this community is mismatched but if you go back to the source information you will see that the information that was sent was uploaded as they were sent. So, it becomes an issue, it is a recurring issue not just for this State but for all the others we have visited so far in the course of trying to move from paper-based report to electronic reporting" (Federal Kwara, scale)*

**3.4.3. Quality of data.** Field diaries provided a qualitative understanding of people's experiences around quality of the data. Major themes around the quality of data were detection of data discrepancies, and concerns expressed with the data entries.

Participants expressed excitement about their ability to quickly detect data incongruency and entry errors which were then followed up and corrected in the system immediately. Such errors included double entry of treatment data, gaps in the drug inventory and community listings. State level participants who regularly monitored the DHIS2 dashboard were able to spot errors in data entries and notify the LGA teams for necessary corrections. In some instances, users of the platform at the LGA levels often reached out to the Command Centre to verify data and ensure entries are error-free. This

was described as a necessary step in ensuring data accuracy and quality. This would encourage teams to be accountable from both ends to pursue good data quality.

*"I have been able to realize that 3 out of the 10 facilities under Sule tankanrkar as at today no data entry coming from 3 wards (Dansoma, Danlaed and Jekie. This will enable me also to give support to Suletankarkar coordinator in other to ask what is going on in these wards and why we have not received information from those wards. Though it is good cause, I've been able to pinpoint exactly where issues are and be able to support the process that led to quality map as well as timely data reporting so I must say that this DHIS is fantastic and yes, it will greatly improve the manner at which we do our work"* (Federal, Jigawa pilot)

*"Today, we found out negative gaps in the drugs/medicine. What I mean is that drugs they reported is in excess of what they received or what they had. What we did was call the LGA, they had to go back to their entries, they found out it was an error when they were entering the data and they corrected the data, and the negative gaps immediately reverse. This is very interesting because as you are seeing any issue on the dashboard on the DHIS2, you are calling immediately, and it was immediately rectified."* (LGA, Enugu scale).

*"We looked at it based on target population and the numbers treated, I looked at the medicine given, what was used, what was wasted, the gap, the ratio and it was fun, easy to spot and make those corrections rather than wait for it to pile up till the end and then you start doing your data cleaning, with the DHIS2 we were able to pool everything, export to excel, analyse the data and then make informed decision"* (Federal, Jigawa scale)

In some of these settings, State level participants reported that data on the dashboard appeared manipulated and largely implausible thus flagging data quality issues to be followed up on.

*"Also in Kaiama, for the 90 communities that have been entered in the platform, data on persons treated and medicine used are very unrealistic …-the drug used and drug received were the same: no loss, no balance! It calls for major doubt. We called the coordinator, and the M&E has promised to look into it …"* (State, Kwara scale).

*"One thing I found out with sule-tankarkar LGA specifically is seeing irregularity with their data entry especially due to the fact that numbers of persons treated is always a round figure with the entries, they are sent and such a way the number of the mectizan used, looks to me to be doctored because they just put a multiplying factor of 2.5 to it. I believe this will require technical supports from the LGA or the supporting partner"* (Federal, Jigawa pilot)

### 3.4.4. Use of DHIS2 data.
The field diaries were utilised as the primary source of data to understand how the DHIS2 data was used in the field as well as some of the barriers to this, although examples of use cases were flagged throughout all qualitative data. The most common uses of the DHIS2 were for identifying areas of low coverage, informing decisions about places to conduct supervisory visits, monitoring data quality and tracking drug inventory and adjustment of MDA activities. Participants in both pilot and scale up States generally regarded the use of the platform as a significant improvement of MDAs compared to previous MDAs without the DHIS2 platform.

*"I was able to see at the command center that Taura recorded huge numbers of IVM brought forward and the figure is the same as what was given to them. So I flagged it off for them to check and ensure data are properly registered under the appropriate column so there seems to be a mix up with drug brought forward and drug received and what was used, so with the platform I was able to flag it up real time and ask the team to look at what they are reporting. Nice!"* (Federal, Jigawa pilot)

### 3.5. Self-administered questionnaire

Assessment of government ownership indicators were measured at three points: before MDA, immediately after training, and after MDA are presented in Table 4 and further summarised for each State in Tables C-E in S1 File.
Overall, participants' scores for government data ownership were significantly higher (P<0.001) across all three states following the rollout of the DHIS2 platform during the most recent MDA, compared to the previous MDA without the DHIS2. The greatest increase in mean ownership scores was observed in Enugu state (28% difference) followed by Kwara Kwara (21%), and Jigawa (10%).

### 3.6. Focus group discussions and key informant interviews

Results from the FGDs and KIIs evaluated participants' understanding of data ownership, barriers to scaling up the platform and what improvements were needed to optimise the platform.

**3.6.1. Improved data ownership.** Participants across the different levels generally considered the platform as a reliable way of monitoring and tracking who and what changes are made to the data by those at the LGA levels as an important aspect of ownership of the system.

*"There is a significant improvement with regards to quick analysis correlation and the reporting of data, quick access, uh, easy tracking of errors, less time consuming. And it helped us significantly in having a one unique and uniform data bank for us to go and access at every time, at any time I need for you to visualize, analyse, and take any proper action or decision-making concerning data. So, it's very important" (KII Auyo coordinator, Jigawa).*

*"Yes, of course, DHIS helps you also to track who, when and what was changed on the platform, so if there is a particular data you see that you are not sure of, you can as well go down to the audit history, then you find out ok, in all States, who did it or what was done, even if it is the State person that changed it, you can as well see it that the person changed it, and you find out what was the issue. So, I think it helps us track who change the file, what was changed and when it was changed" (FGD FMoH.)*

Participants were excited about the platform being a reliable replacement of the old paper-based system that was prone to losses and errors. Participants also believed that having this single source of treatment data that could be accessed centrally by all the levels of the health system ultimately increased ownership at all levels especially among persons at the LGAs as they were confident that what they have entered onto the DHIS2 platform is the same as that available to all NGDO partners. Another perceived benefit of the system was the shorter time it takes to collate data for reporting, as participants did not need to travel to collect data or request access from NGDO partners.

*"It is very much easier now unlike before when you have to carry bundle of papers up on date just to get a particular data. But now, even on my personal tablet, I can access the data" (KII Igbo-Etiti M&E, Enugu).*

**Table 4. Comparison between participants expected ownership score versus what was obtained in reality.**

| State | Expectation (Pre-Intervention: last MDA) | Reality (Post-Intervention: post MDA) | Difference in ownership scores (Pre versus post-MDA) |
|---|---|---|---|
| Enugu | 64% (29.0/45) | 93% (41.7/45) | **28%*** |
| Jigawa | 86% (38.9/45) | 96% (43.3/45) | **10%*** |
| Kwara | 73% (32.8/ 45) | 94% (42.4/45) | **21%*** |

*\* P<0.001*

*"Remember that before now, the partners will treat, State will treat, collect data, the hardcopy or State will treat, partner will go to the LGA to collect data without involving the State. But now the State will be in the office and be seeing all the information they need, so whether the partner goes to the LGA to collect hardcopy is immaterial - it doesn't really matter. So, they now own their program, they have access to their program, they have access to their outcome, they make decisions on their own with the outcomes. So, ownership has really improved on the side of the owners of the information" (FGD FMoH).*

The ability to view data entries after submission was perceived by LGA level participants as a way of strengthening ownership, as opposed to when data submitted to the FMoH was out of their influence or reach.

*"After recording on the system, I can still refer to it anytime I want to. Apart from that, it has also helped me and my M&E to work cooperatively on the entry of data" (KII data manager, Kwara).*

**3.6.2. Barriers *to* Scaling *up the* DHIS2 platform.** Barriers to scale was also an emerging theme during the FGD and KIIs with Federal, state and LGA level implementers. Participants identified shortages of human resource, high attrition rate of programme staff, inadequate funding, technical expertise and poor network infrastructure as the major barriers to scaling up the platform to other settings. Results are summarised in Table 5.

## 4. Discussion

The key components of ownership (access, control, use) and those of our primary research question (access, use, quality) overlapped significantly and findings were often related to scale. Our analysis of the platform's metadata revealed that both state and LGA-level actors had real-time access to and actively used the data, marking a significant departure from previous MDAs where access to the data was only possible at the end of the MDA campaign. This shift underscores the transformative role of the DHIS2 platform in enhancing data accessibility and aligns with previous findings by Zerfu et al. (2024), who though not focussed on NTDs, documented improvements in accessing health data across primary health-care facilities in Ethiopia [17].

The field diaries and process evaluation both revealed that while access to the platform was generally reported as good, internet availability was identified as a major factor affecting successful implementation of the platform, with quality or cost being the major factor impeding accessibility. This is similar to findings from Bangladesh and Kenya where internet connection were highlighted as major impediment to deploying and adopting the DHIS2 platform [18,19]. Mentioned across the States and across all levels of the administrative structure was the assertion that internet availability physically prevented the system from working properly, led to a drain on time and reduced data quality. The issues of technology,

**Table 5. Barriers to scaling up the DHIS2 platform as highlighted by participants.**

| Highlighted barriers | Source: KII | Source: FGD |
|---|---|---|
| Lack of adequate human resources to scale-up of the DHIS2 platform nationally for MDA reporting | State & LGA levels | Federal Level |
| Inadequate funding to procure equipment such as laptops, internet set-up etc hinders scale up especially in States without a supporting NGDO partner for NTD programme | State & LGA levels | Federal Level, State level |
| Poor internet infrastructure especially in rural settings threatens access and use of the DHIS2 for scale up | State level | Federal Level |
| Inadequate technical competence in use of DHIS2 at the lower level of the health system. | LGA level | Federal Level |
| High rate of staff attrition from the NTD programme to other programmes within the MoH thus necessitating the need for constant retraining of new programme staff posted to the NTD units both at the Federal and State NTD programmes. | – | Federal Level |

infrastructure and network are similar to those reported in another study exploring the acceptability of the DHIS2 platform for NTDs in Cameroon and remain critical to successful implementation of the platform [20]. It is therefore important that these barriers be sufficiently planned for and addressed in the scale out plan, to achieve both data ownership and data quality.

A major theme from the field diaries around data quality was the positive correlation between data access, data quality and use. DHIS2 dashboard has previously been reported as valuable for enhancing monitoring and quality of programmatic data [21], this aligns with our findings, where participants from both the State and Federal levels reported being able to visualize data through the dashboard and spot errors in the data - including duplicate entries, incorrect populations, errors with treatment and medicine numbers - and were also able to get them checked and addressed. Similarly, those at State levels, reported occasions where they suspected the data had been manipulated and flagged them for investigation. This fostering of accountability further highlights the role of the DHIS2 platform in enhancing data quality.

The observed discordance in treatment coverage estimates between the DHIS2 system and the CES contradicts earlier studies that noted a 72% concordance rate between routinely reported and surveyed coverage across 12 African countries [22]. While some variation is expected, it was particularly concerning that coverage estimates from the DHIS2 platform in the scale-up States fell outside the confidence intervals of CES results. The different denominators used by each data source may likely explain this observation. For example, coverage from DHIS2 was estimated using administrative target population as denominator, which may not reflect real-time population dynamics such as migration, urbanisation, or seasonal mobility. On the other hand, CES uses sampled population data, which better captures current demographic realities but may also introduce sampling bias depending on methodology [23,24]. Moreover, it is also likely that the CES may be sampling different population subgroups than those reached during MDA and reported through the DHIS2 system, especially in areas with high population mobility. This finding highlights the need for programme managers to triangulate data from different sources when making planning decisions based on coverage estimates.

Participants reported various uses of the data from the platform, primarily for tracking MDA performance in real-time, planning for upcoming MDA, and informing decision making. The ability to use data more extensively than before was perceived as a meaningful improvement, contributing to a sense of optimism among participants. The DHIS2 platform also enabled real-time monitoring of MDA performance, allowing for programme adjustments as needed, and improving the agility of the MDA process. Positive perceptions of data utilisation were consistently linked to improved decision making, better coverage tracking, and more effective medicine inventory management. During the review meetings, participants highlighted the platforms role in fostering collaborative review of the data, sharing of experiences for tackling challenges and planning of upcoming MDA campaigns. This clear use case of the DHIS2 platform for enhancing decision making and fostering collaboration across heath system, has been noted by several studies and underscores its potential for transforming data into actionable insights. This clear use case of the DHIS2 platform for enhancing decision making and fostering collaboration across heath system, has been noted by several studies [25,26], and underscores its potential for transforming data into actionable insights.

We observed an overall positive trend in perceived data ownership among participants, which was often linked with data accessibility across all the levels. The existence of one *"uniform data bank"* as one participant called it, fostered an increased sense of program ownership between the different administrative levels and the NGDO partners. Furthermore, real-time data access and ability to track changes to that data empowered MDA teams with a sense of control over the data. Increased ownership was further evidenced by improved efficiencies, particularly around time saved not needing to aggregate data at each level and the ability of teams at each level to address issues promptly instead of waiting until after the campaign where it may be too late to rectify. Both LGA and State levels shared experiences of increased collaboration with other local teams, such as Logistics and Monitoring & Evaluation. The enthusiasm and sustained interest from the FMoH bode well for scale up and their motivation to take ownership of the system, assuming full leadership and technical independence from NGDO partners.

Our study showed that there is currently high enthusiasm at the Federal level to scale use of the system nationally. There are however perceived challenges associated with scaling up the platform that the FMoH will need to focus on - notably the additional workload required from the LGA cadre who are saddled with most of the access and data entry tasks, capacity building needs, and frustration with poor connectivity, as highlighted by previously studies. Motivation will therefore be key to scale-up. In additional to this, NGDO partners will need to be lobbied to support the model: as it is not guaranteed that all NGDO partners will be as supportive as the ones engaged in this research. Many of the barriers highlighted here cut across States but also appear to be mediated in part by the technical partner and the location or context. Furthermore, each State and NGDO partner have different operating models, cultural contexts, existing technology capacity and levels of willingness to change. It will therefore be contingent upon the FMoH to take into account that successful scale-up into new locations will not be a one size fits all approach. Additionally, our research focuses only on the Nigerian specific context. Further studies in diverse settings could help provide a broader understanding on how to best replicate and scale-up the platform, to improve data ownership, access, use, and quality in different MDA implementation contexts. Further considerations for scaling up the tool are presented by themes as a series of recommendations in Table 6.

## 5. Conclusions

This is the first time the already established DHIS2 platform for reporting other public health conditions has been adapted for real-time monitoring and reporting of MDAs in Nigeria. Our study shows the interconnected nature of data ownership, access, use, and quality in the context of a data management platform for MDA programmes. With the platform enhancing

**Table 6. Recommendations for scaling up the tool.**

| Themes | Recommendations for consideration in scale-up |
|---|---|
| LGA workload | The FMoH needs to create a clear validation process for the spreadsheets used at the LGA level to be replaced by DHIS2. The removal of this Excel sheet will be integral to successful scale-up considering the reduction in the associated workload at the LGA level. |
| Community lists | Better and earlier engagement on the community listings should be initiated ahead of the MDA, including a mechanism for LGAs to request community additions or removals moving forward. |
| Connectivity | Pro-active hardware and infrastructure assessments will be needed at LGA and State levels to ensure that minimum requirements are in place or can be put in place prior to roll-out. |
| Capacity building | Training will be key to the success of this platform. The FMoH and State levels will need an increased focus on data quality and use (isolating both MDA and data entry issues), while the LGA level will need increased focus on platform interaction, especially on data visualisation. |
| Establishing a clear plan for handover to the FMoH | The handover plan needs to be clearly mapped out and agreed upon by all parties. One important component of this will be capacity building at the FMoH – on the process, configuration and platform management – creating a large group of experts to take this forward. |
| NGDO meeting led by FMoH | If the FMoH is planning a national scale-up of this MDA reporting platform, then NGDO (implementing partner) involvement is indispensable. The scale-up plan and budgetary implications using the platform (Command Centre and increased communication costs) need to be outlined in detail. |
| Support for timely State-level data review meetings | The data review meetings were extremely popular, and multiple requests that they are retained were recorded in the meeting notes. Although these meetings are part of the programme more generally and not specific to DHIS2 scale-up, the immediate availability of the data makes these meetings more feasible and of a higher utility. |
| Clearer guidance on roles and process | As this platform is standardised, terms of reference for each level need to be thoroughly documented and well communicated during each training, as well as during other MDA meetings such as planning or data reviews. |
| Separate drug tracking platform | A separate drug supply chain dashboard is needed on the platform, with additional data sources included so a full picture of drug use and availability at each level can be viewed by the logistics teams at Federal, State and LGA levels. |
| Enhancements to the platform | Additional functionality with the aim of improving the quality of data entered can be set up in DHIS2, like introducing minimum and maximum ranges for certain indicators and using the data validation platform, making sure certain logical criteria are met within the data. |

data access, quality, and utilisation at all levels of the health system, so did the perception of data ownership among participants. The study provides valuable insights for scaling up the program nationally, both within Nigeria and other endemic countries looking to emulate their success in monitoring MDA treatment in real-time. While the platform demonstrated ability to be replicated in different settings, the study acknowledges the need to consider different operating models, cultural contexts, and technical capacities across the diverse locations. Notably, addressing technology and infrastructure limitations, ensuring data quality, and fostering engagement with NGDO partners will be crucial to the successful scale-up and sustainability of this process.

## 6. Limitation

We acknowledge several limitations in this study that should be considered when interpreting the findings. First, the selection of States was guided by implementation readiness and existing partner support structures. While this approach facilitated smooth deployment of the DHIS2 system, it limits the generalisability of the findings to other States or within the broader national contexts. Second, the study captured only one MDA round per state (even in Jigawa MDAs were implemented for different diseases in the pilot and rollout). As such, it does not fully capture insights into the sustainability or consistency of DHIS2 integration over multiple MDA cycles, which is critical for long-term programmatic success. Finally, the study largely relied on subjective methods; a more objective assessment would be beneficial.

## Supporting information

**S1 File. The supplemental file includes a detailed description of the responsibilities of each implementing teams before, during and after the MDA (Table A), data collection tools such as the ownership statement, which was used in assessing various components of data access, use, and control (Table B); as well as the questions for the online pool (Table C).** It also contains assessment results from different states regarding ownership measurements (Tables D, E & F), and participants' responses presented in word clouds. These word clouds highlight the improvements and challenges related to data access, use, and quality based on a rapid survey conducted with research participants (Fig A).
(DOCX)

## Acknowledgments

We wish to thank our study participants from the different states, as well as partners from the Carter Centre, HANDS and CBM for their support. We also acknowledge the contribution of Matthieu Chevallier, who supported the production of audio-video training materials for data collectors to conduct the remote Coverage Evaluation Survey (CES) during the COVID-19 lockdown.

## Author contributions

**Conceptualization:** Martins Imhansoloeva, Sunday Isiyaku, Ruth Dixon, Sarah Bartlett.

**Data curation:** Martins Imhansoloeva, Christian Nwosu, Omosefe Osinoiki, Chukwuma Anyaike, Perpetua Amodu-Agbi, Augustine Nwoye, Peter Oyinloye, Amin Umar Abdurahman, Ifeoma Otiji, Lazarus Nweke, Joseph Kumbur, Rinpan Ishaya, Ruth Dixon, Sarah Bartlett.

**Formal analysis:** Martins Imhansoloeva, Christian Nwosu, Ruth Dixon, Sarah Bartlett.

**Funding acquisition:** Martins Imhansoloeva, Sunday Isiyaku, Chukwuma Anyaike, Ruth Dixon, Sarah Bartlett.

**Investigation:** Martins Imhansoloeva, Christian Nwosu, Omosefe Osinoiki, Sunday Isiyaku, Chukwuma Anyaike, Perpetua Amodu-Agbi, Ununumah Egbelu, Augustine Nwoye, Amin Umar Abdurahman, Ifeoma Otiji, Lazarus Nweke, Joseph Kumbur, Rinpan Ishaya, Ruth Dixon, Sarah Bartlett.

**Methodology:** Martins Imhansoloeva, Christian Nwosu, Sunday Isiyaku, Perpetua Amodu-Agbi, Ununumah Egbelu, Augustine Nwoye, Ruth Dixon, Sarah Bartlett.

**Project administration:** Martins Imhansoloeva, Christian Nwosu, Omosefe Osinoiki, Sunday Isiyaku, Perpetua Amodu-Agbi, Ununumah Egbelu, Augustine Nwoye, Ruth Dixon, Sarah Bartlett.

**Resources:** Perpetua Amodu-Agbi, Lazarus Nweke, Joseph Kumbur, Rinpan Ishaya.

**Supervision:** Martins Imhansoloeva, Omosefe Osinoiki, Perpetua Amodu-Agbi, Ununumah Egbelu, Augustine Nwoye, Peter Oyinloye, Ruth Dixon, Sarah Bartlett.

**Validation:** Ununumah Egbelu, Ruth Dixon, Sarah Bartlett.

**Visualization:** Christian Nwosu, Perpetua Amodu-Agbi, Ununumah Egbelu, Augustine Nwoye, Peter Oyinloye, Amin Umar Abdurahman, Ifeoma Otiji, Lazarus Nweke, Joseph Kumbur, Rinpan Ishaya, Ruth Dixon, Sarah Bartlett.

**Writing – original draft:** Martins Imhansoloeva, Omosefe Osinoiki, Ruth Dixon, Sarah Bartlett.

**Writing – review & editing:** Martins Imhansoloeva, Christian Nwosu, Omosefe Osinoiki, Sunday Isiyaku, Chukwuma Anyaike, Perpetua Amodu-Agbi, Ununumah Egbelu, Augustine Nwoye, Peter Oyinloye, Amin Umar Abdurahman, Ifeoma Otiji, Lazarus Nweke, Joseph Kumbur, Rinpan Ishaya, Ruth Dixon, Sarah Bartlett.

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
