## [Decision Letter · Decision Letter 0]

12 May 2025

PNTD-D-25-00416

Evaluating DHIS2 as a real-time treatment and stock reporting tool for mass drug administration in Nigeria

Dear Dr. Imhansoloeva Martins,

Thank you for submitting your manuscript to PLOS Neglected Tropical Diseases. After careful consideration, we feel that it has merit but does not fully meet PLOS Neglected Tropical Diseases's publication criteria as it currently stands. Therefore, we invite you to submit a revised version of the manuscript that addresses the points raised during the review process.

Please submit your revised manuscript within 60 days, due on Friday July 18, 2025. If you will need more time than this to complete your revisions, please reply to this message or contact the journal office at plosntds@plos.org. Please include the following items when submitting your revised manuscript:

We look forward to receiving your revised manuscript.

Kind regards,

Grace Adira Murilla, PhD

Academic Editor

Amy Morrison

Section Editor

Shaden Kamhawi

co-Editor-in-Chief

Paul Brindley

co-Editor-in-Chief

**Additional Editor Comments:**

Reviews provide very constructive suggestions that would increase the impact of the manuscript.  Please address carefully consider all reviewer suggestions or a clear rational if you don't.  In particular providing clear hypotheses/objectives suggested by reviewer #2 and strengthening the qualitative analysis by inclusion of statistics should be considered.

Thanks.

**Journal Requirements:**

At this stage, the following Authors/Authors require contributions: Martins Imhansoloeva, Christian Nwosu, Omosefe Osinoiki, Sunday Isiyaku, Chukwuma Anyaike, Perpetua Amodu-Agbi, Ununumah Egbelu, Augustine Nwoye, Peter Oyinloye, Amin Umar Abdurahman, Ifeoma Otiji, Lazarus Nweke, Joseph Kumbur, Rinpan Ishaya, Ruth Dixon, and Sarah Bartlett. Please ensure that the full contributions of each author are acknowledged in the "Add/Edit/Remove Authors" section of our submission form.

- ® on page: 8.

4) Please amend your detailed Financial Disclosure statement. This is published with the article. It must therefore be completed in full sentences and contain the exact wording you wish to be published. Please ensure that the funders and grant numbers match between the Financial Disclosure field and the Funding Information tab in your submission form. Note that the funders must be provided in the same order in both places as well.

**Reviewers' Comments:**

Reviewer's Responses to Questions

**Key Review Criteria Required for Acceptance?**

**Methods:**

-Are the objectives of the study clearly articulated with a clear testable hypothesis stated?

-Is the study design appropriate to address the stated objectives?

-Is the population clearly described and appropriate for the hypothesis being tested?

-Is the sample size sufficient to ensure adequate power to address the hypothesis being tested?

-Were correct statistical analysis used to support conclusions?

-Are there concerns about ethical or regulatory requirements being met?

Reviewer #1: 1) Introduction scope – The opening section is clear and instructive, but it presents preventive‑chemotherapy (PC) NTDs as an uncontested approach. Consider adding a short balanced paragraph (2–3 sentences) outlining key arguments for (programme efficiency, integrated platforms) and against (risk of systematic non‑compliance, limited impact on transmission for some NTDs) large‑scale PC‑MDAs, with citations, and which parts of the MDA does the DHIS2 approach aim to monitor and optimise, and the one it does not (e.g., systematic non-compliance).

I would also add "for" or "against" after "PC" in line 74.

2) Methods organisation – The “Platform and process” to "After the MDA campaign" subsections are dense and largely narrative, perhaps more suited for the methods than for introduction. Also, to save some text, I suggest to convert it into a figure/flowchart (e.g. swim‑lane diagram) showing data flow: CDD → LGA focal person → State → national DHIS2 instance, including helpline feedback loops.

3) Table 2 ambiguity – The table headings do not make clear whether each row’s number is (e.g., MDA administrators, surveyed households). Later the households become the denominator for coverage verification, but “NA” appears in the same column. If applicable, i suggest splitting the information: (i) sample characteristics of MDA staff (if relevant) and (ii) household survey frame. Clearly state who each number represents and ensure consistency. Moreover, Table 2 seems best suited for the start of the results section, particularly in the first sentence of the results.

4) The rationale for choosing the three States within Nigeria and the two different partner‑support models is mentioned, but the criteria are not spelled out. Please add, if reasoning available, a brief paragraph explaining why these particular settings were considered representative (and what variation they capture).

5) The quantitative analysis is described in three paragraphs (lines 296‑313) with no mention of significance threshold (95%CI?), clustering adjustments (if any). To note that the coverage surveys interview individuals nested within households, which are themselves nested within communities/LGAs. If you analyse the raw person‑level data as though every observation were independent, the standard errors (and thus confidence intervals and p‑values) will be too small, because responses from the same household or community are correlated. Clustered standard errors may correct for this by telling the software that observations share a common error structure within each cluster (e.g. household ID/community-ID). In Stata this is as simple as adding , vce(cluster hhid) to a regression or proportion command.

6) Because DHIS2 instances can contain personally identifiable information, consider clarifying whether any personal data (names, phone numbers) were stored on the study server and, if so, how they were protected.

Reviewer #2: The manuscript lacks a sentence specifically stated the primary and secondary objectives. Stating a measurable primary hypothesis would strenghthen the focus. A mixed-methods, pre-post comparison is suitable. However, most of the data are qualitative and subjective indicator are used. Objective pre- vs. post-metrics (such as coverage, stock accuracy, timeliness) that the introduction highlights are large absent or anecdotal. The DHIS2 system was piloted and rolled out in three states. The rationale why the study states were selected remains unclear. The sample counts are reported for surveys, yet no power or saturation justifications is provided. Descriptive statistics are shown; however, no inferential tests are applied to pre-post comparisons, which may weaken the conclusions about improvements by leaving them without statistical support. In terms of ethical concerns, the authors state that Nigeria's National Health Research Ethics Committee granted a waiver. No overt ethical issues are evident.

Reviewer #3: (No Response)

**Results:**

-Does the analysis presented match the analysis plan?

-Are the results clearly and completely presented?

-Are the figures (Tables, Images) of sufficient quality for clarity?

Reviewer #1: 7) In Figure 1, Put x‑axis label (“State”) and y‑axis label (“Number of trained users who accessed DHIS2”) directly on the graphic. Mention the figure at its first logical point (in line ~336: “93 % in Enugu…”).

8) Coverage inconsistency analysis – The discussion notes mismatches between DHIS2 and coverage survey but stops short of analysing why. Could this discrepancies stem from numerator inflation/deflation, denominator mis‑estimation, or delayed data synchronisation. This could make recommendations more actionable, as it is one of the major challenges of MDA's monitoring programmes.

9) Table 3 text - Consider briefly stating the main categories of issues if relevant (stock‑outs, duplicate records, etc.) and give one illustrative example of resolution per category. A brief paragraph interpreting the pattern (e.g. why drug‑inventory issues dominated in Kwara) would help the reader; otherwise the table remains descriptive. 

10) Large negative value in Table 4 – The Enugu‑Isi Uzo difference (−21.85 pp) appears very large. Re‑check the calculation, and in all tables fix numbers to one decimal place for consistency (or other place, as authors prefer). Consider adding numerator/denominator of coverages in a supplementary appendix, as this can be relevant to understand the importance of difference in coverage between DHIS2 and coverage surveys.

11) Table 5 formatting Explain why some rows are bold (e.g. p < 0.05) in a footnote.

Reviewer #2: The analysis partially matches the analysis plan. The manuscripts outlines a mixed-methods evaluation. The analysis of MDA treatment and stocking reporting is largelly based on users' perception, which is somewhat subjective. Few quantitative estimates are applied to demonstrate the effect of DHIS2 platform on improvement. Objective indicators such as drug stock trakcing, inaccurate coverage reportings. This may limit the strength of the conclusion.

The results are moderately clearly and completely presented. The qualitative results are clearly structured by theme and supported with quotations. Yet, more analytical techniques, such as code-code cooccurrence and thematic analysis should be employed to deepen the interpretation. If possible, the authors should use objective estimates for key outcomes--such as treatment coverage, drug usage, and stock wastage, etc--to strengthen the validity fo their findings and reduce reliance on self-reported or perceived improvements.

The quality and clarity of figures and tables need improvement. Although the tables and figures are informative, there are occasional inconsistencies or lack of clarity. Some figures would benefit from reformmating.

Reviewer #3: (No Response)

**Conclusions:**

-Are the conclusions supported by the data presented?

-Are the limitations of analysis clearly described?

-Do the authors discuss how these data can be helpful to advance our understanding of the topic under study?

-Is public health relevance addressed?

Reviewer #1: The paper provides one of the first empirical assessments of DHIS2 for real‑time MDA stock and treatment reporting in an African setting. With the clarifications above, it will serve as a useful operational blueprint for other NTD programmes and digital‑health implementers.

12) Definition of “treated” – Although informative, a minor limitation of the study regarding coverage would be the definition of "swallowed" as this was self-reported and may be subject to desirability bias. Consider also other potential biases and limitations, such as Selection bias – States were purposively chosen; results may not extrapolate nationally; Short follow‑up – The study captures one MDA round per State; sustainability over multiple cycles is untested; Incomplete cost analysis – No costing of data bundles, helpline staff time or device purchase; important for scale‑up.

13) The manuscript notes that Jigawa out‑performed other States on ownership scores, but does not explore whether partner intensity (pilot study), prior DHIS2 exposure or baseline capacity contributed to this difference.

Reviewer #2: The conclusions are partially supported by the data presented. The manuscript effectively summarizes user-reported improvements in data access, usage, perceived ownership. Yet many of these conclusions are based on qualitative feedback without corresponding objective measures. The limitations of the analysis--particularly the reliance on subjective data and lack of inferential test/interpretive analysi-- should be more explicitly discussed.

The public health relevance is clearly addressed, as the study is positioned within efforts to improve programmatic efficiency and disease control through better data system. Clarifying the connection between improved data processes and their impact on health outcomes would further reinforce the public health significance of the study.

Reviewer #3: (No Response)

**Editorial and Data Presentation Modifications?**

Reviewer #1: 14) Possible redundancy – 9 main tables + 1 figure and abundant descriptions crowd the methodology and results section; some may be best suited to supplementary material. Consider moving some tables to supplementary appendix; while Retaining main results in the text and refer readers to the appendix.

15) Table 1 - Add an abbreviations footnote (e.g. NW, SE, SW, MDA, LGA, CBM, etc.).

16) Section numbering - Introduce hierarchical numbering (e.g. 3 Results, 3.1 DHIS2 meta‑data, 3.2 Helpline monitoring, 3.3 Treatment coverage data, 3.4 field diaries, 3.4.1 accessibility of data) to help navigation.

17) The manuscript is generally well written. However, as mentioned before, consider trimming lengthy methodological/results detail and moving it to Supplementary Methods. The manuscript currently reads half scientific article, half programme report.

18) correct "meticizan" in line 489 to "mectizan"; "log shows" to "log showed" in line 351;

19) a couple of more references in the discussion could be beneficial.

Reviewer #2: Several tables and figures require reformatting to improve clarity and ensure consistency throughout the manuscript.

Reviewer #3: (No Response)

**Summary and General Comments:**

Reviewer #1: The study by Imhansoloeva et al. tackles a high‑priority operational challenge for national neglected‑tropical‑disease (NTD) programmes: obtaining timely, reliable treatment‑coverage and drug‑stock data during mass‑drug administration (MDA). The mixed‑methods design (DHIS2 metadata analytics, helpline monitoring, treatment‑coverage verification and field diaries) is appropriate and, in places, impressively granular. Nevertheless, some clarifications and streamlining steps are needed before the manuscript is ready for publication.

Reviewer #2: This manuscript addresses a timely and relevant topic: the use of the DHIS2 platform to support mass drug administration (MDA) for neglected tropical diseases (NTDs) in high-disease burden settings. The research is grounded in a solid background, with a clear public health rationale and alignment with global NTD elimination goals. The integration of both qualitative and descriptive quantitative data provide useful insights into user experience, system adoption and perceived improvements in data accessibility, quality and owenership, etc.

However, the presentation of the data requires improvement. Greater use of structured formatting, clearer legends, and visual enhancements would enhance clarity and impact. In addition, there is also a gap between the study's stated objectives and the data actually analyzed. These outcomes are primarily discussed through qualitative user perceptions, with limited use of objective estimates. While the qualitative findings are well-organized and supported by quotations, the analysis would benefit from deeper qualitative analytical teniques. There is no concern regarding research ethics.

Reviewer #3: (No Response)

PLOS authors have the option to publish the peer review history of their article (what does this mean? ). If published, this will include your full peer review and any attached files.

**Do you want your identity to be public for this peer review?** For information about this choice, including consent withdrawal, please see our Privacy Policy .

Reviewer #1: **Yes: ** Luís-Jorge Amaral

Reviewer #2: No

Reviewer #3: No

**Figure resubmission:**
---

## [Editor Report · Decision Letter 1]

24 Sep 2025

Dear Dr. Imhansoloeva_Martins,

We are pleased to inform you that your manuscript 'Lessons from piloting and scaling a real-time DHIS2 based treatment reporting tool for mass drug administration in Nigeria' has been provisionally accepted for publication in PLOS Neglected Tropical Diseases.

Best regards,

Grace Adira Murilla, PhD

Academic Editor

Amy Morrison

Section Editor

Shaden Kamhawi

co-Editor-in-Chief

Paul Brindley

co-Editor-in-Chief

The manuscript is accepted after the following minor comments have been addressed.

Minor Comments

1. Figures: Quality is low. Please ensure adherence to PLOSNTD Journal standards.

Figure 1 appears to have two titles. This needs to be corrected.

In the same Fig. 1, within the "Command Post" box, the text "Watch Reports for gaps in data" should be replaced with "Review Reports for gaps in data" Data cannot be watched.

2. The titles of all Figures and Tables must be descriptive. Ensure that all abbreviations used are defined, including those used in Tables

Figure 2: "Proportions of persons accessing the DHIS2 platform by States and administrative level". This title is incomplete. It does not indicate where the study was carried out. FMoH appears in the figure but not defined in the legend.

Authors should ensure the manuscript quality meets the expected standards.

4. Grammar check should cover the whole manuscript including texts in the flowchart.

Thank you.

---

## [Editor Report · Acceptance letter]

Dear Imhansoloeva,

We are delighted to inform you that your manuscript, "Lessons from piloting and scaling a real-time DHIS2 based treatment reporting tool for mass drug administration in Nigeria," has been formally accepted for publication in PLOS Neglected Tropical Diseases.

Best regards,

Shaden Kamhawi

co-Editor-in-Chief

Paul Brindley

co-Editor-in-Chief
